# Water Immersion Aging of Epoxy Resin and Fique Fabric Composites: Dynamic–Mechanical and Morphological Analysis

**DOI:** 10.3390/polym14173650

**Published:** 2022-09-02

**Authors:** Michelle Souza Oliveira, Fernanda Santos da Luz, Artur Camposo Pereira, Ulisses Oliveira Costa, Wendell Bruno Almeida Bezerra, Juliana dos Santos Carneiro da Cunha, Henry Alonso Colorado Lopera, Sergio Neves Monteiro

**Affiliations:** 1Department of Materials Science, Military Institute of Engineering-IME, Praça General Tibúrcio 80, Urca, Rio de Janeiro 222290-270, Brazil; 2CCComposites Laboratory, Universidad de Antioquia UdeA, Calle 70 No. 52-21, Medellin 050010, Colombia

**Keywords:** natural fiber, fique fabric, epoxy resin, composite, aging, water immersion, dynamic–mechanical analysis

## Abstract

Fiber-reinforced composites are among the most investigated and industrially applied materials. Many studies on these composites using fibers, especially with natural fibers, were made in response to an urgent action for ambient preservation. A particularly relevant situation exists nowadays in the area of materials durability. In this respect, no studies on water-immersion-accelerated aging in fique fiber–epoxy composites are reported. This work aimed to fill this gap by investigating the epoxy matrix composites reinforced with 40 vol% fique fabric. The epoxy matrix and the composite, both unaged and aged, were characterized by weight variation, water absorption, morphology, colorimetry (CIELAB method), Fourier transform infrared spectroscopy (FTIR) and dynamic–mechanical analysis (DMA). The main results were that degradation by water presents appearance of complex microfibril structures, plasticization of epoxy resin, and debonding of the fique fiber/epoxy matrix. The most intense color change was obtained for the water-immersion-aged epoxy by 1440 h. Cole–Cole diagrams revealed the heterogeneity of the materials studied.

## 1. Introduction

In recent years, a rapid growth in the use of polymeric composites reinforced with fibers occurred, producing a combination of high performance, versatility and advantages at favorable costs [1]. In this respect, natural fibers have become an important class of reinforcement materials, having characteristics that are of great interest in the area of polymeric composites, due to their low density, low cost, biodegradability, flexibility in processing and their renewable and non-toxic sources. The mechanical properties of these composites might also be significantly improved, as in the exemplary case of toughness of flax fiber-reinforced epoxy matrix composites added with silica nanoparticles [2]. However, the hydrophilic characteristic of a natural fiber hinders not only its adhesion to the hydrophobic polymer matrix but may also affect the durability of the composite in the case of prolonged contact with water. Water absorption by composite materials exposed to the environment, after a long period in use, is considered an unavoidable phenomenon and can result in matrix plasticization, structural damage and crack growth [3,4]. Green composites are prone to moisture absorption in a humid environment or when immersed in water. In these cases, moisture diffusion can be described by three mechanisms. First, moisture diffuses through pores and cracks in the polymer matrix. Second, capillary transport occurs between the fiber/matrix interface. Ultimately, natural fibers swell after absorbing moisture, and this leads to microcracks in the matrix around the swollen fibers. Eventually, this leads to a permanent debonding of the matrix and fiber because water-soluble substances leach from the fiber surface [5,6]. The hygroscopic behavior of natural fibers influences the biodegradation characteristic of the material. Moreover, greater moisture absorption facilitates microbial attack. All these phenomena affect the interfacial adhesion between fiber and matrix, which leads to poor stress transfer and changes the physical, mechanical and thermal properties of the composite [5,7,8]. The moisture diffusivity in the composite depends mainly on time, temperature, environment, matrix properties, curing technique, and fiber orientation, among others [9]. It is known that water can affect not only the polymer matrix but also the fibers and even the interface, causing loss of adhesion, thus influencing the integrity of the composite. The polymer matrix absorbs water by a diffusion process, in which case the kinetics of absorption generally follow Fick’s law [4,10,11,12,13].

Haameem et al. [14] suggested that moisture diffusion can be briefly categorized into three mechanisms. The first mechanism is associated with the diffusion of water molecules within the microgaps between the polymer chains. In the second, water molecules are transported by capillary to the gaps and gaps between the fibers, as well as to the adhesive bond of the matrix. Finally, in the third mechanism, microcracks occur in the composites due to the swelling of the natural fibers. In general, moisture diffusion in a composite depends on certain factors such as fiber volume fraction, void content, matrix viscosity, moisture and temperature. Additionally, other factors that may influence mechanical properties are the size of the natural fiber (short or long), porosity, relative humidity and manufacturing method.

The dynamic thermomechanical analysis (DMA) of polymeric matrix materials has been applied in the development of materials to evaluate mechanical stability [7,15], thermal limit [16,17] and sound isolation [18]. In combination with the damping energy determination, which is associated with molecular motions displayed by a polymer in solid state, this analysis can be applied to the study of morphology in multiphase systems. DMA tests are mainly considered to study modifications in the composite, with the incorporation of fiber and the performance as a function of properties. The current literature presents ample findings in this field [10,16,17,19,20,21]. However, there is an absence of DMA studies on water-immersion-accelerated aging in natural fiber–epoxy composites. Fique fibers are quite a unique natural fiber occurring mainly in Colombia [22,23,24,25,26,27], which are of regional importance and valid for other natural fibers. Therefore, this work investigated for the first time the water immersion aging epoxy matrix composites reinforced with 40 vol% fique fabric. The epoxy matrix and the composite were characterized by weight variation, water absorption, scanning electron microscopy on the aged surface, structure of individual molecules and the composition of molecular mixtures considered using Fourier transform infrared spectroscopy, as well as dynamic mechanical analysis.

## 2. Materials and Methods

### 2.1. Materials

The fique fabric, Figure 1, was purchased in the market of Medellin, Colombia. The development of techniques for spinning and weaving the natural fibers results in the production of composites with superior mechanical properties [28]. The fabrics are manufactured by interlacing the weft yarns (0°) with the warp yarns (90°) in a simple and regular arrangement. The bidirectional fabric areal density, i.e., 859 g/cm${}^{2}$, was measured according to the corresponding standards in a previous article [22,29]. Table 1 summarizes the main properties of fique fibers.

The matrix phase consisted of the diglycidyl ether of bisphenol A (DGEBA) epoxy resin mixed with the triethylenetetramine (TETA) hardener in the stoichiometric proportion of 100/13 (DGEBA/TETA). Both DGEBA and TETA were supplied by Epoxyfiber, Rio de Janeiro, Brazil.

### 2.2. Fabrication of Fique Fabric–Epoxy Matrix Composites

The fique fabric pieces were placed in an oven at 70 °C for at least 24 h. The 40 vol% fique fabric–epoxy matrix composite was fabricated by placing the previously dried fabric inside a steel mold while pouring fluid DGEBA-TETA resin in a layer by layer manner. Laminated plates of 150 × 120 × 3.2 mm were prepared by the compression molding process and cured inside the mold under a 5-ton load at room temperature (RT) for 24 h. For the DMA tests, these plates were cut using a jigsaw model BOSCH Professional GST 75 E into specimens with dimensions of 50 × 13 × 3.2 mm. After manufacturing, these specimens were subjected to accelerated water absorption aging in the process described as follows.

### 2.3. Water Absorption

Absorption tests were performed on dried rectangle specimens of both the neat epoxy resin and the composite. The tests were carried out in a deionized water bath (closed container) at RT (25 °C), according to ASTM D570-98 [30]. Long-term water absorption was conducted by immersing the specimens for different time periods, of up to 144 days, aiming to study their durability. After immersion for a determined time, the specimens were taken out and thoroughly dried with a clean cloth. All specimens were weighted again. The difference of weight between the sample in dry condition and that after water immersion time *t* was obtained by:(1)$$\%WA=\frac{w_{final}-w_{initial}}{w_{initial}}$$
where *w${}_{final}$* is the weight of the sample at a specific time period of water immersion and *w${}_{initial}$* is the weight of the dry sample at *t* = 0. The content of absorbed water in composites after immersion time *t* (*M${}_{t}$*) was calculated by the weight difference between the samples immersed in water and the dry composite samples. The water absorption kinetic model [31,32,33] was used to describe the sorption curves, as:(2)$$\frac{M_{t}}{M_{\infty }}=kt^{n}$$
where *M${}_{t}$* and *M${}_{\infty }$* are water content at time *t* and at equilibrium; *k* and *n* are constants which provide information about the diffusion mechanism that is acting in the composites. The diffusion capacity of composites, which is represented by the ability of the water molecules to move inside the specimens, was evaluated by the water diffusion coefficient (*D*) via the following equation:(3)$$D=\pi {\left(\frac{kh}{4M_{\infty }}\right)}^{2}$$

### 2.4. Accelerated Aging

The water-immersion-aged samples were conditioned at RT in a plastic box and stayed submerged during the entire aging period. Three groups were tested corresponding to aging of 10, 20 [34] and 30 days [35] of exposure. Table 2 presents the specimen’s nomenclature.

### 2.5. Scanning Electron Microscopy Analysis

Samples from the fique fabric, neat epoxy and composites aged by immersion in water were analyzed by scanning electron microscopy (SEM) in a model Quanta FEG250, FEI. To carry out these analyses, a metallic coating was applied on the surface of the samples using the LEICA equipment, model EM ACE600.

### 2.6. Colorimetry Analyses: CIELAB Color Space

The CIELAB color space, Figure 2, is a second method for providing a perceptually uniform color space. In this color space, the distance between two points approximately tells how different the colors are in luminance, chroma and hue. Colorimetric analysis was performed to assess surface color changes, using a chromometer, and evaluate the results using the coordinates of this method. The *L${}^{*}$* represents the brightness value, as in the darkest black *L${}^{*}$* = 0 and in the brightest white *L${}^{*}$* = 100. The components green–red and blue–yellow are represented by the chromaticity coordinates *a${}^{*}$* and *b${}^{*}$*, respectively.

The red and yellow components are shown in the positive direction, while the green and blue components are in the negative direction. The total color changes (Δ*E*) were calculated as described in ISO 7724-1 [36] according to:(4)$$\Delta E=\sqrt{{\left(L_{2}^{*}-L_{1}^{*}\right)}^{2}+{\left(a_{2}^{*}-a_{1}^{*}\right)}^{2}+{\left(b_{2}^{*}-b_{1}^{*}\right)}^{2}}$$
where *L*, *a* and *b* are the differences between the initial values (unweathered sample (1)) and final (weathered sample (2)) of *L${}^{*}$*, *a${}^{*}$* and *b${}^{*}$*. The whiteness index, Equation (Equation 5), is a numeric indicator used as the degree of whiteness. In the CIELAB color space, two of the axes are perceptibly orthogonal to the luminosity. The hue can be calculated along with the chroma, Equation (Equation 6), transforming coordinates a and b from rectangular form to polar form. The hue is the angular component of the polar representation, while chroma is the radial angular component.
(5)$$W_{i}=100-\sqrt{{\left(100-L\right)}^{2}+a^{2}+b^{2}}$$
(6)$$C_{ab}^{*}=\sqrt{\left(a^{2}+b^{2}\right)}$$

The colorimetric pattern evaluation of the plates was performed in a colorimeter portable model WR-10QC of the brand ARTBULL, CN.

### 2.7. Fourier Transform Infrared Spectroscopy

Fourier transform infrared spectroscopy (FTIR) was performed to identify and determine functional groups of composite and epoxy structures in the infrared (IR) region between 400 cm${}^{-1}$ and 4000 cm${}^{-1}$. The absorption spectra were obtained with a resolution of 4 cm${}^{-1}$ and 64 scans in each assay. The equipment used was a Thermo Scientific spectrometer, using the OMNIC Spectra software.

### 2.8. Dynamic–Mechanical Analysis (DMA)

DMA, also known as thermodynamic-mechanical, was performed to obtain information on the viscoelastic behavior, such as the storage modulus (E${}^{\prime }$), loss modulus (E${}^{\prime \prime }$), the ratio between these moduli, i.e., tan δ, as well as the glass transition temperature (T${}_{g}$) of the samples, described in Table 2. The procedure was carried out following the ASTM D4065 [37] standard in the three-point bending mode. The samples dimensions were 55 × 13 × 3 mm. The equipment used was a DMA Q800 from TA Instruments. The test parameters were: amplitude of 20 μm, frequency of 1 Hz, static force of 2 N, range of heating from 30 to 190 °C and heating rate of 3 °C/min in a nitrogen atmosphere.

## 3. Results and Discussion

### 3.1. Water Absorption

The moisture diffusion in the natural fiber composites is influenced by the volume fraction of fiber [32], as well as the humidity, the voids, the viscosity of the matrix and the temperature [38,39]. It is well-known that the water absorption capacity depends upon the constituents of natural fibers and might reveal how their composites behave when moisture is present in practical applications, either in a total immersion or partial water exposure situations. Figure 3 shows the water absorption and diffusion coefficients graphs, as well as the *n* and *k* parameters, for both the epoxy and the fique fabric/epoxy composite. One should note that there is a large increase in the water diffusion process when the epoxy resin (0.006 × $10^{-4}$) is compared with the fique-fabric-reinforced composite (3.044 × $10^{-4}$), as well as a significant increase in water absorption values in the composites. These results are expected since natural fibers are hydrophilic materials and, in this case, absorb a considerable volume of water since they are the composite’s reinforcement with 40 vol%. Meanwhile, porosity and microvoids in the epoxy matrix might also have contributed to the resulting water absorbed. The variation in moisture content can cause a network of microfractures on the surface of the epoxy and the composite.

### 3.2. Weight Variation

An initial increase in weight was observed, as shown in Figure 4. It was mainly associated with the composite water diffusion process. This event was followed by a second increase in weight that reaches a maximum value, around 23%. Exposure for 1440 h recorded a weight variation of the order of 20%, indicating that the saturation for the composite is around 20%, since there was no significant change in the percentage of weight variation. As mentioned before, natural fibers typically have a high moisture absorption capacity, which can lead to fiber swelling and loss of dimensional stability. In addition, swollen fibers decrease the fiber/matrix adhesion as well as the mechanical properties of their composites [40]. Regarding the epoxy matrix, the hydrolytic degradation process in the polymer is characterized by a decrease in the crosslink density, an increase in the hydrophilicity of the network and leaching of low molecular weight products [41]. Krauklis et al. [41] point out three types of leaching that are potentially possible: (i) hardener leaching; (ii) leaching of epoxy compounds; and (iii) leaching of impurities or additives. In theory, there is a possibility that some amount of unlinked hardener, in the case it is water soluble, would be washed out of the crosslinked polymer network or used in further crosslinking.

### 3.3. Effects of Water Immersion Aging on Fique Fabric

Figure 5 shows the evolution of the degradation in the fique fabric due to water immersion. Through SEM images, the appearance of complex microfibril structures is noticed as the aging process proceeds. Excessive water absorption leads to an increase in bound-absorbed water and a decrease in free water. In this situation, water can penetrate the cellulose network of the fiber as well as in the capillaries and spaces between the fibrils and less connected areas. Additionally, water can also form chemical bonds with groups on cellulose molecules. The rigidity of the cellulose structure is destroyed by the water molecules in that structure, in which moisture acts as a plasticizer, allowing the cellulose molecules to move freely. Hence, the cellulose mass is softened and can change the fiber dimensions easily with an application of force [42,43]. Wei and Meyer [44] observed in their study that fibers with an intact surface, which did not receive any surface treatment, present better resistance to degradation, while fibers with surface defects are prone to high precipitation of calcium hydroxide in the cell walls, when applied in a cement matrix. The authors’ conclude that internal lignin and hemicellulose easily undergo alkaline hydrolysis, which further leads to the separation of the cellulose microfibrils. Gómez Hoyos and Vázquez [34] observed the same structure present in this study for the fibers after alkaline treatment. The authors inferred that the removal of hemicellulose and lignin made the interfibrillar regions less dense and rigid, allowing the fibrils to reorganize along the direction of tension, resulting in better charge sharing.

### 3.4. Effects of Water Immersion Aging on Epoxy

Figure 6 presents the effects of water immersion degradation on the epoxy resin. The SEM images show that the epoxy surface was markedly damaged, consequently, microvessels, infiltration species through which water can be percolated, are easily observable. In addition, a slight yellowing of the samples was observed, as well as an increase in the amount of pores. The nature of the interaction of sorbed water with epoxy has long been debated in the literature [8,41,45,46,47,48]. Although it is well-accepted that the sorbed water plasticizes the resin, resulting in a change in the glass transition temperature, modulus and shear strength of the polymer, the exact mechanism of the adsorbed water/epoxy interaction is not fully understood. A schematic diagram was proposed by Panchagnula et al. [47] to represent the nature of the water/epoxy interaction in which water molecules are shown not only to be hydrogen-bonded to the epoxy but also hydrogen-bonded to themselves, resulting in the formation of free interstitial water.

### 3.5. Effects of Water Immersion Aging on Fique-Fabric-Reinforced Epoxy Composite

Figure 7 shows that one of the main events that water caused in composites is the fiber–epoxy debonding. It might have occurred due to the aforementioned swelling of the fique fiber. As a result, microcracking of the brittle epoxy matrix occurs [42]. The high cellulose content in the natural fiber further contributes to more water penetrating the interface through the microcracks induced by fiber swelling and creating stresses that lead to composite failure [42]. As the composite cracks and is damaged, mechanisms of capillarity and transport through microcracks become active. The capillarity mechanism involves the flow of water molecules along the fique fiber/epoxy matrix interfaces and a diffusion process through the matrix. Water molecules actively degrade the interface, resulting in debonding between the fique fiber and the matrix [42,49]. Diffusion can occur due to capillary action and is predominant in composites where fiber wettability by the matrix is incomplete. The transport behavior of matrix voids and the fiber/matrix interface might have a substantial effect on the overall diffusivity of composite materials [50]. Induced moisture has detrimental effects on the performance of composite materials, leading to matrix plasticization, chemical and mechanical degradation [6,7,8,48].

### 3.6. Colorimetry Analysis: CIELAB

It is important to have an objective way to characterize colors of polymers and their composites which were subjected to water absorption, as well as to quantify differences between colors. Considering that, the CIELAB analysis was of paramount importance. Figure 8a shows an increase in *L${}^{*}$* and *b${}^{*}$* parameters as water exposure increase for epoxy resin. The parameter *a${}^{*}$* was stable, i.e., without significant changes. Regarding the composite, a greater variation in the parameters was observed, but when compared with the unaged composite and to the most severe exposure condition (AC/1440 h), no significant change was found in the intensity of *L${}^{*}$*, *a${}^{*}$* or *b${}^{*}$*. Figure 8b shows that the most intense color change was obtained for the AE/1440 h, also increasing the whiteness index. This is due to a more whitish film observed on the epoxy surface after exposure to water. The initial uncured epoxy resin, and even the yellowish TETA, could explain the yellowing of the epoxy resin and its composite over time. Residual crosslinking can also cause a decrease in the concentration of the unreacted amine group causing the color change [41].

### 3.7. Fourier Transform Infrared Spectroscopy

Figure 9 shows the FTIR spectra of the epoxy and composite samples, both unaged and under more severe aging conditions (1440 h).

Regarding the epoxy (Figure 9a), the band displacement can be seen at 560 cm${}^{-1}$, 1491 cm${}^{-1}$ and 1614 cm${}^{-1}$. Significant increases in absorbance bands at 818 cm${}^{-1}$ and 2354 cm${}^{-1}$, and decreases at 943 cm${}^{-1}$, associated with the stretching C-O of the oxirane group, 1158 cm${}^{-1}$, 1326 cm${}^{-1}$ and 1570 cm${}^{-1}$, associated with the bending vibration of the primary amino group, were observed. Cañavate et al. [51] reported that the bands close to 863 and 917 cm${}^{-1}$ own the assignment of the epoxy and anhydrous ring, and these bands are known to be intensity-decreased during the curing reaction due to the opening of the epoxy rings. Therefore, both are used as an indication for the progression of the curing reaction. In contrast to this, and unexpectedly, the band at 3419 cm${}^{-1}$ was greatly reduced. This band is attributed to the O-H stretching of the hydroxyl groups (3650–3600 cm${}^{-1}$ (free) and 3400–3200 cm${}^{-1}$ (hydrogen bond), which is commonly increased due to the hydrolysis reaction. Ether bonds (C-O ⇒ 1300–1000 cm${}^{-1}$) are the most sensitive bonds to hydrolysis in the epoxy network and explain the lack of band at 3400 cm${}^{-1}$ [45]. This band is known to be related to O-H stretching [47,52]. The Blackburn et al. [53] studies indicated increases in the degree of polymerization throughout the 8-week exposure period despite variations in moisture and temperature conditioning. This occurs based on the fact that the epoxy was initially cured under laboratory conditions. Regarding the composite (Figure 9b), a shift and increase was observed in the band 580 cm${}^{-1}$ and an increase at 1486 cm${}^{-1}$, associated with the symmetrical stretching CH${}_{2}$, as well as a significant decrease in the absorbance bands at 723 cm${}^{-1}$, 890 cm${}^{-1}$ and 1796 cm${}^{-1}$. Reductions in the bands 810 cm${}^{-1}$ and 1063 cm${}^{-1}$, associated with the C-O-C ether elongation, and 1264 cm${}^{-1}$ were noted, as well as the appearance of the band in 2345 cm${}^{-1}$. According to Ray and Rathore [50], the chemical degradation of the composite includes the hydrolysis of the bond at the epoxy/fiber interface.

### 3.8. Effects of Water Immersion Aging on DMA

One of the main objectives of DMA is to relate macroscopic properties to molecular relaxations associated with conformational changes and microscopic deformations generated from molecular rearrangements. The DMA technique separates the dynamic modulus (E) of the material into two distinct parts: an elastic (storage) (E${}^{\prime }$) and a viscous (loss) (E${}^{\prime \prime }$) components. The ratio of E${}^{\prime \prime }$ to E${}^{\prime }$ (E${}^{\prime \prime }$/E${}^{\prime }$) gives the tangent of the phase angle δ, tan δ, which is known as the damping and may be regarded as a measure of the energy dissipation capacity of the material. By using the curve of tan δ, the glass transition temperature (T${}_{g}$) can be obtained, which may also be used to evaluate the material characteristics. The DMA curves in the α-transition zone of the epoxy and composite aged by water immersion are shown in sequence.

#### 3.8.1. Storage Modulus (E${}^{\prime }$)

Figure 10 shows the storage modulus values (E${}^{\prime }$) vs. temperature curves for each investigated material. One can see that AC/1440 h had a decrease in the E${}^{\prime }$, which could be due to the softening and the beginning of the relaxation processes within the polymer matrix [15]. Arias et al. [45] also observed that the storage modulus and the glass transition values decreased with hydrolysis. These -OH bridges have been reported to be the reason for the more stable storage modulus of epoxy, as observed at AE/240 h and AE/720 h. Regarding the composite, the AC/240 h showed a significant drop (lower E${}^{\prime }$) due to the damaged matrix, deteriorated interfacial adhesion and bond strength between the matrix and the fiber. Chemical combination behaviors of the chains, van der Waals bonding and hydrogen bonding in polymer molecular construction were responsible for the material’s ability to carry the external stress. Once water molecules entered the epoxy, the hydrolysis and plasticization of the matrix damage the chemical combinations and bondings. Experiencing a stress may induce greater strains that could lead to an E${}^{\prime }$ decrease. The E${}^{\prime }$ of AC/720 h and AC/1440 h was higher when compared with the UC.

#### 3.8.2. Loss Modulus (E${}^{\prime \prime }$)

Figure 11 shows the variation of E${}^{\prime \prime }$ with temperature for the investigated epoxy resin and its composite.

An increase in the loss modulus is observed in all cases, i.e., the increase in viscosity with the aging process. It is suggested that the shifts to higher temperatures are caused by a decrease in the molecular movement. Additionally, with increasing time exposure to water, the intensity of E${}^{\prime \prime }$ peaks gradually increases and becomes broader for both the epoxy resin and its composite. This behavior reveals that the aging effectively suppresses the polymeric chains mobility, resulting in a broadening of the T${}_{g}$ range. Such behavior is related to the degradation processes such as plasticization, residual stress relaxation, hydrolysis and so forth [19,54]. Despite this, based on the presented results, it is not possible to differentiate these phenomena and individually evaluate their influence on the degradation process.

#### 3.8.3. Tangent Delta (tan δ)

The tan δ or damping curve of the epoxy resin and its composite after different saturation times in the water are presented in Figure 12. One may notice that the damping value decreased with the prolonged water immersion time, indicating higher elasticity [20,45,46]. The intensity of the tan δ peaks for the immersed composites decreased to around half of the untreated sample values. Arias et al. [45] observed the existence of two peaks, indicating bimodal chain lengths. In the present study, two peaks were also observed after aging for AE/720 h, becoming more apparent after 1440 h of exposure. As for the composite, the same behavior was observed, and all curves presented bimodal or more chain lengths.

The T${}_{g}$ related to the observed peaks of tan δ showed a slight decrease based on the water exposure, i.e., there is an increase in the temperature at which the transition from a glassy state occurs (in which the molecules of the amorphous phase do not have mobility) to a rubbery state (when the amorphous phase molecules become mobile). This phenomenon possibly occurs due to the presence of water molecules bound to the epoxy network. Incontestably, the value of T${}_{g}$ was the most affected property in this process due to the existence of secondary -OH groups in the network [45], which are also associated with epoxy degradation. Water became a bridge between the -OH groups in the epoxy network and as a lubricant when in excess, producing greater mobility of these chains. All aged conditions showed an increased T${}_{g}$ when compared with the UC. After 1440 h, the T${}_{g}$ had a reduction of 4% in comparison with that achieved after aging for 720 h, this was also observed by Uthaman et al. [46]. The authors reported that the T${}_{g}$ of the composites first increased and then was similar to the control group samples, remaining stable under all submersion conditions. Concerning this final reduction, the authors attributed it to the plasticization effect on the materials submerged for a long period of time. This effect is generally defined as the increase in the segmentation mobility of the polymeric chains. However, an increase in the restriction of the movement of the epoxy molecules was observed based on the decrease in the peak in tan δ. Energy dissipation may have been increased due to increased voids in the epoxy.

### 3.9. Qualitative Discussion

Table 3 presents the main parameters obtained from the E${}^{\prime }$, E${}^{\prime \prime }$ and tan δ curves in Figure 10, Figure 11 and Figure 12, respectively. The results in this table reveal a significant effect of water immersion aging on important DMA parameters of the fique-fabric-reinforced epoxy composites. With up to 720 h, the epoxy resin storage modulus was increased by more than 81% with respect to the UE. On the other hand, E${}^{\prime }$ decreased by 30% after 1440 h of water exposure. Regarding the composite, the storage modulus was increased by more than 73% compared with the UC. As for the loss modulus, the AE/1440 h was heightened by more than 160% as compared with the UE. Regarding the composite, the loss modulus was elevated more than 90% to AC/1440 h as compared with the UC.

The maximum internal friction associated with the peak in E${}^{\prime \prime }$ was slightly affected by water immersion aging. Indeed, the internal friction of either AE/240 h or AC/240 h in Table 3 was more than 15% greater than that of the UE and UC, respectively. In contrast, the maximum damping associated with the peak in tan δ decreased, while the dynamic T${}_{g}$ temperature at the peak increased with the exposure time to the water immersion aging. The transition temperature from the glassy to the rubbery conditions in Table 3 was significantly increased. This suggests that water immersion aging interfered in the thermally activated mechanisms that promote the glassy to rubbery transition.

### 3.10. The Cole–Cole Plot

The viscoelastic parameters E${}^{\prime }$ and E${}^{\prime \prime }$ can also be represented in the Cole–Cole plot, where E${}^{\prime \prime }$ is plotted as a function of E${}^{\prime }$ on the linear axes, as shown in Figure 13. Cole–Cole plots provide information about the secondary relaxations, structural changes after filler addition [20] and system heterogeneity. Homogeneous polymeric systems exhibit semicircular curves, while heterogeneous multiphase systems present imperfect or elliptical curves [21]. The imperfect semicircular shapes have been associated with the relatively good interfacial adhesion at fiber loading [16]. Figure 13 shows the Cole–Cole diagrams for the UE and UC and their aged treatments.

The Cole–Cole curves in this figure show that the UE and aged epoxy conditions are homogeneous systems with a concave shape (semicircles). However, one can see an increased widening effect on the curves for the conditions with exposure times of up to 720 h, which result in higher final E${}^{\prime }$ values. For the samples with 1440 h of exposure, a reduction in amplitude and a decrease in final E${}^{\prime }$ were noticed, as well as the beginning of an “irregularity”, i.e., the formation of two semicircles. This irregularity has been related to two different relaxation mechanisms which correspond to the secondary relaxations [15]. All conditions presented higher E${}^{\prime \prime }$ in comparison with the unaged epoxy. Regarding the UC and aged composite treatments, the Cole–Cole diagram shows only one semicircle to the UC and two semicircles for the other conditions. Additionally, an increase in the values of the E${}^{\prime }$ and E${}^{\prime \prime }$ was observed after the aging treatments. These results show that the aging process effectively suppresses the polymeric chains mobility and is indicative of materials’ heterogeneity associated with greater differences in relaxation processes of the epoxy matrix when exposure continues. Moreover, the Cole–Cole diagram in Figure 13b shows that the 1440 h exposure treatment resulted in the highest E${}^{\prime }$ and E${}^{\prime \prime }$ values. Therefore, it can be inferred that CA/1140 h can absorb higher external loads while exhibiting elastic behavior and maintaining its characteristics at higher temperatures [15].

## 4. Summary and Conclusions

Dynamic mechanical analysis was conducted in both epoxy resin and 40 vol% fique-fabric-reinforced epoxy matrix composites. Data obtained after accelerated water immersion aging were analyzed thoroughly in order to understand the relationship between long-term performance properties and various physico-mechanical parameters. The following conclusions can be drawn from the above study:The analysis of water absorption showed a large increase in water diffusion coefficient when comparing the epoxy resin and the fique fabric composites. The weight variation did not show a clear connection between the amount of water absorbed and the duration of exposure. However, a general trend in the obtained values suggests an increase in the weight with the increase in exposure duration.The SEM analyses showed the deteriorated structure and, in particular, that the interface may initiate complex microfibril structures, microvessels, pores and microcracking of the epoxy matrix due to fiber swelling or fique fiber/epoxy matrix debonding. Consequently, this may additionally influence the composite’s behavior and/or structure integrity. The process of water immersion aging, especially for the epoxy resin, was dominated by residual crosslinking interfering with the color change in the sample.Based on the specific reductions in the absorbance bands (943 and 810 cm${}^{-1}$) of the epoxy and composite observed, respectively, it can be inferred that the curing process occurred through exposure by water immersion.The dynamic moduli behavior confirms the degradation of the investigated epoxy resin and composite during accelerated aging. The E${}^{\prime }$ decreased and the E${}^{\prime \prime }$ increased during continued water exposure. The penetration of water molecules inside the composite’s structure caused the plasticization of the epoxy matrix and initiated the degradation process, possibly by breaking the hydrogen bondings. This behavior is related to the degradation process, which decreases the strength of a structure and increases its viscosity and damping under elevated temperatures. After the accelerated aging, all three aging conditions show considerably higher T${}_{g}$ values for both materials, which is attributed to residual crosslinking.

Therefore, it can be stated that DMA experiments on aged materials can give valuable indications about their long-term performance properties. The reported results can be helpful both on design and operation stages for epoxy-reinforced fique fabric composites when in contact with water.

## Figures and Tables

**Figure 1 polymers-14-03650-f001:**
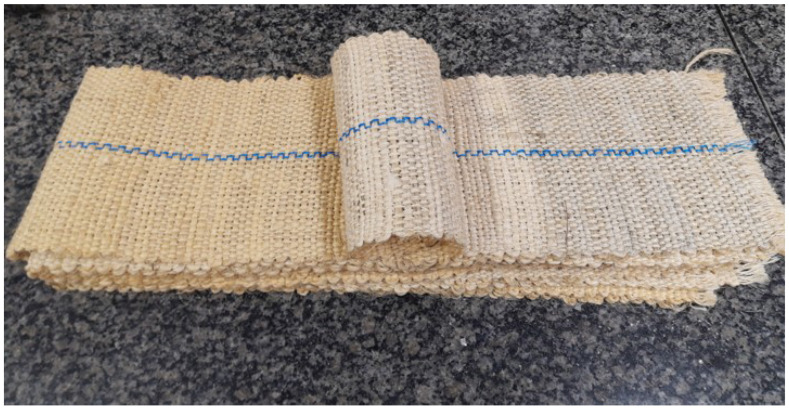
Fique fabric reinforcement.

**Figure 2 polymers-14-03650-f002:**
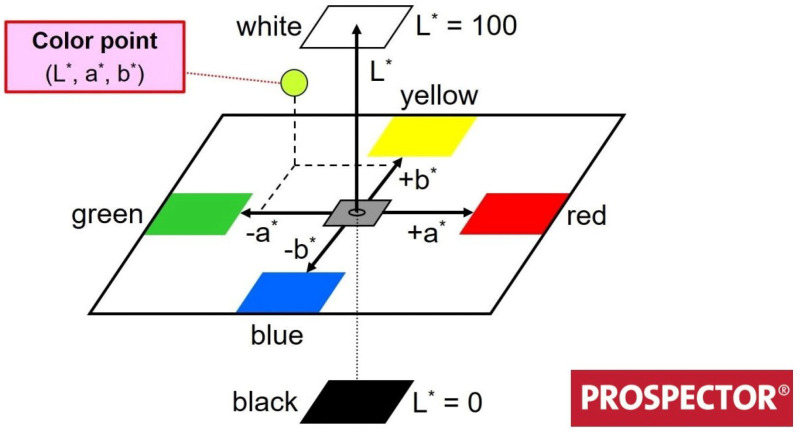
Three-dimensional CIELAB color space.

**Figure 3 polymers-14-03650-f003:**
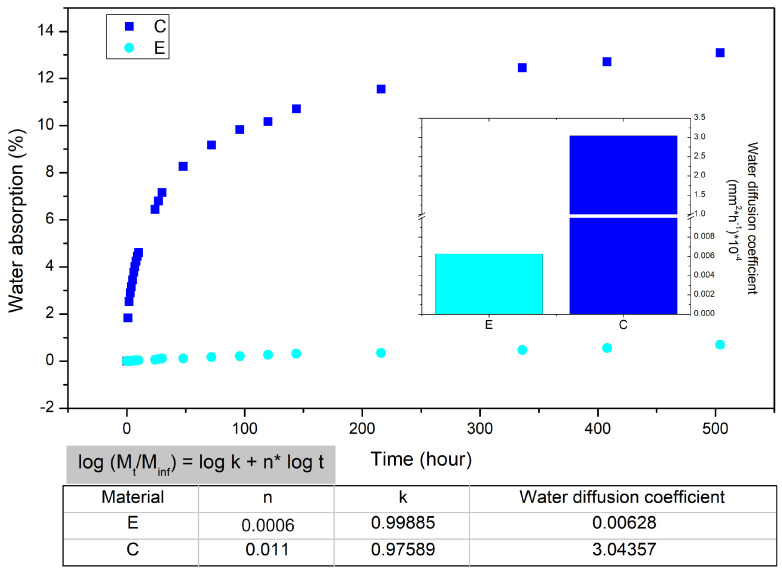
Weight variation of epoxy (E) and fique fabric–reinforced epoxy matrix composite (C) during water immersion aging.

**Figure 4 polymers-14-03650-f004:**
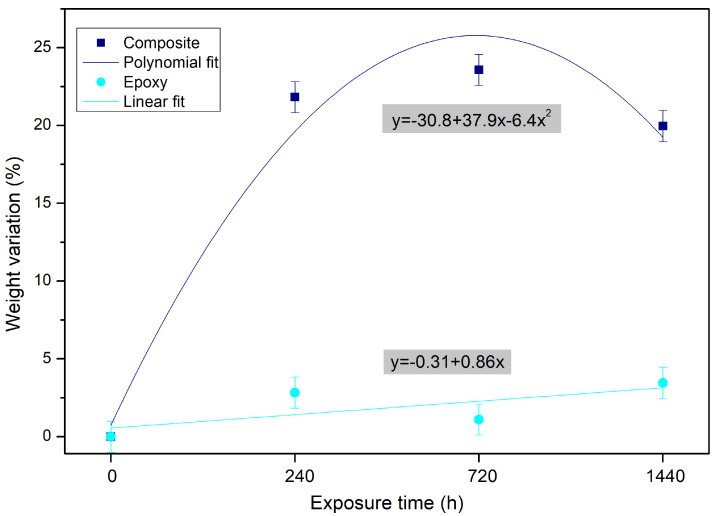
Weight variation of epoxy and fique fabric–reinforced epoxy matrix composite during water immersion aging.

**Figure 5 polymers-14-03650-f005:**
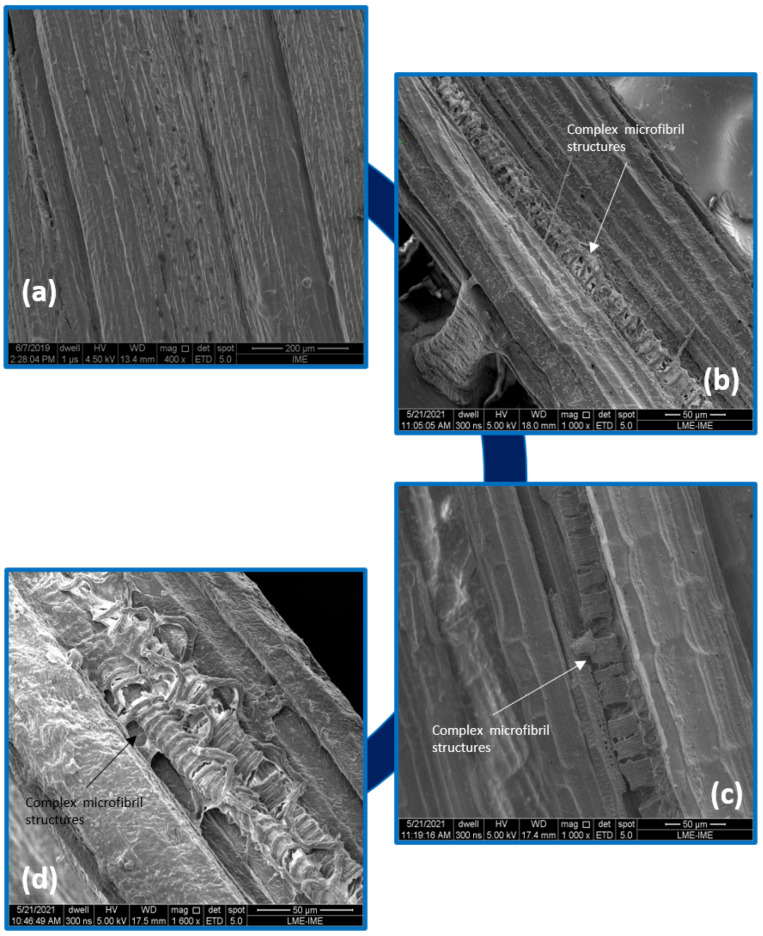
SEM microscopy appearance of the fique fabric: control group (**a**) and water immersion aged after 240 h (**b**), 720 h (**c**) and 1440 h (**d**).

**Figure 6 polymers-14-03650-f006:**
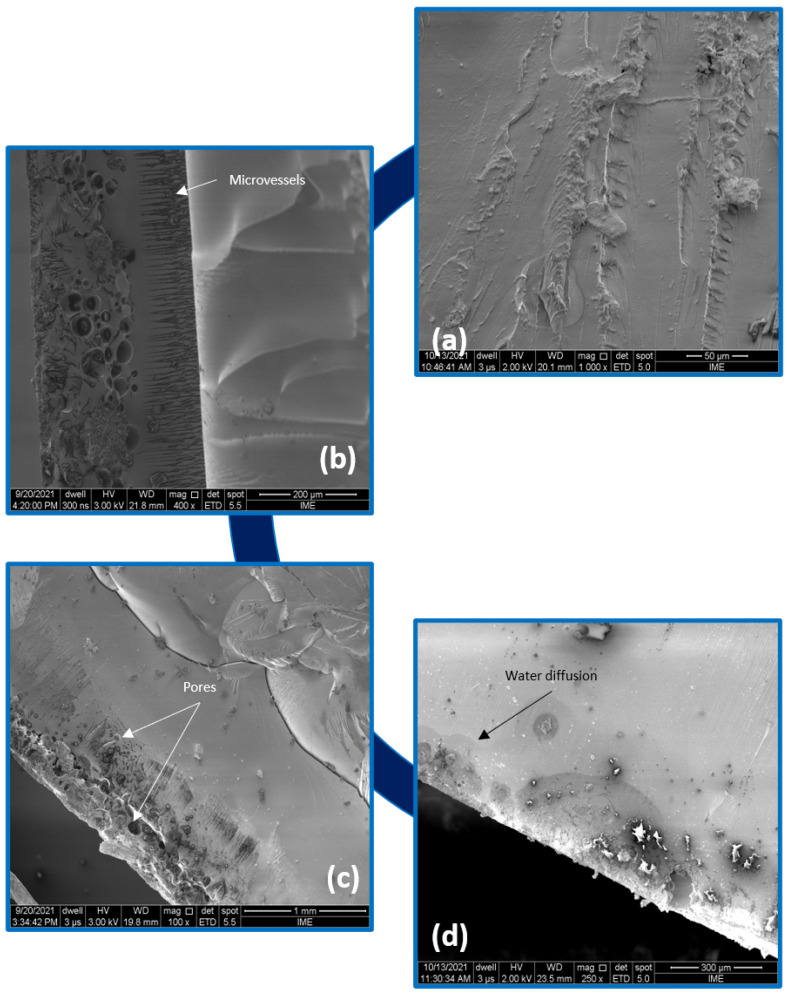
SEM microscopy appearance of the epoxy: control group (**a**) and water immersion aged after 240 h (**b**), 720 h (**c**) and 1440 h (**d**).

**Figure 7 polymers-14-03650-f007:**
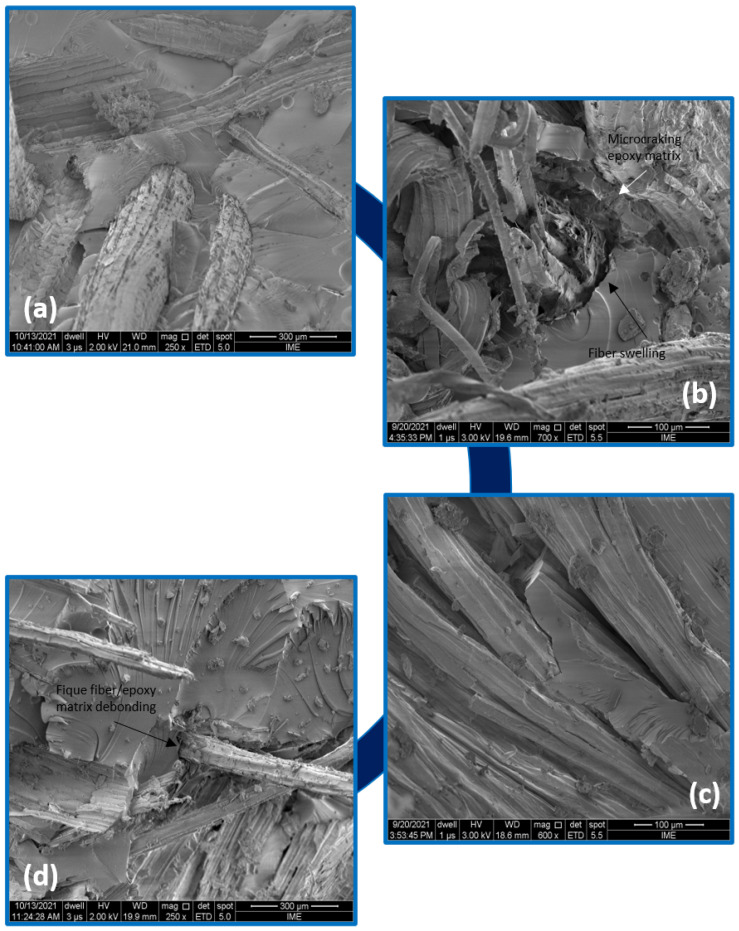
SEM microscopy appearance of the composite: control group (**a**) and water immersion aged after 240 h (**b**), 720 h (**c**) and 1440 h (**d**).

**Figure 8 polymers-14-03650-f008:**
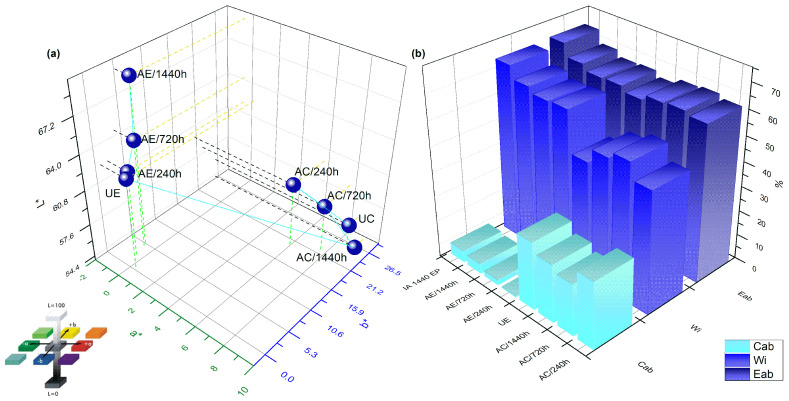
(**a**) Changes in color coordinates of the color space CIELAB for epoxy resin, composite and their (**b**) main parameters during water immersion aging.

**Figure 9 polymers-14-03650-f009:**
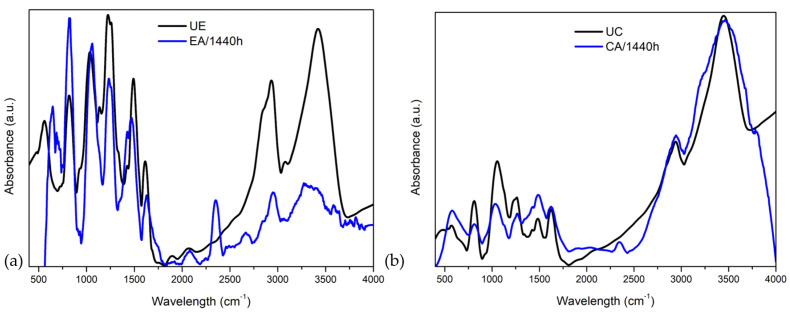
Fourier transform infrared spectra of (**a**) epoxy and (**b**) composite unaged and water immersion aged for 1440 h.

**Figure 10 polymers-14-03650-f010:**
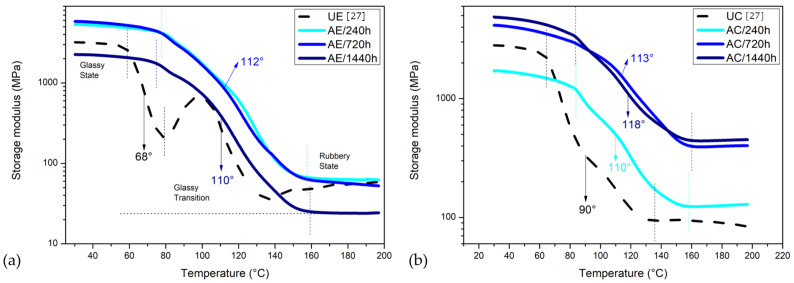
DMA storage modulus (E${}^{\prime }$) curves for (**a**) epoxy resin and its composite unaged and water-immersion-aged (**b**).

**Figure 11 polymers-14-03650-f011:**
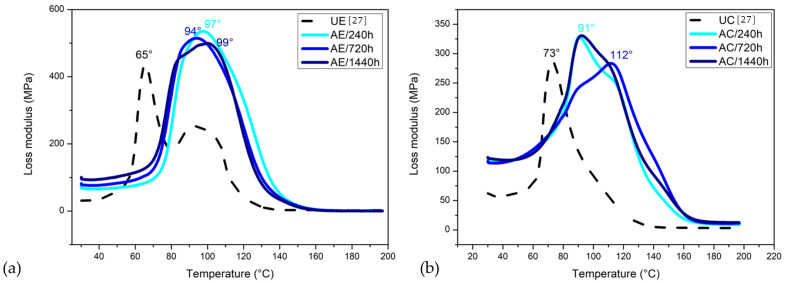
DMA loss modulus (E${}^{\prime \prime }$) curves for (**a**) epoxy resin and its composite unaged and water-immersion-aged (**b**).

**Figure 12 polymers-14-03650-f012:**
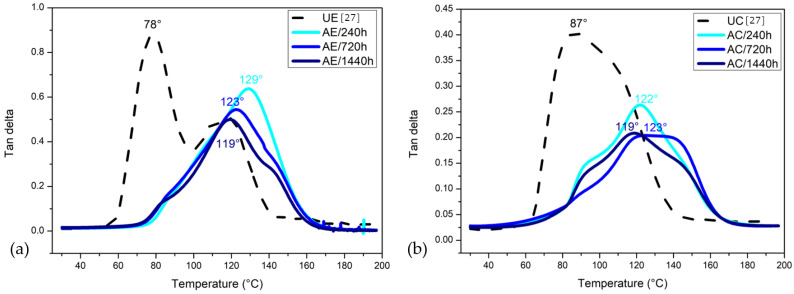
DMA tangent δ curves for (**a**) epoxy resin and its composite unaged and water-immersion-aged (**b**).

**Figure 13 polymers-14-03650-f013:**
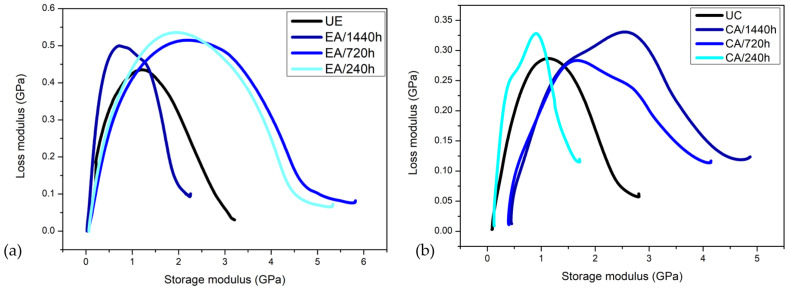
Cole–Cole plots of (**a**) the UE aged epoxy treatments and (**b**) the UC aged composite treatments.

**Table 1 polymers-14-03650-t001:** Chemical and mechanical properties of fique fibers [22,23,24,25,26,27].

**Fique**	**Cellulose** **(wt%)**	**Hemicellulose** **(wt%)**	**Lignin** **(wt%)**	**Wax** **(mm)**	**Pectin** **(wt%)**	**Ash** **(wt%)**
	18.7–70	22.1–27.1	6.81–16.6	-	-	-
	**Microfibril** **Angle (°)**	**Apparently** **Density** **(g/cm^3^)**	**Real** **Density** **(g/cm^3^)**	**Fiber** **Diameter** **(mm)**	**Tensile** **Strength** **(MPa)**	**Elongation** **at Break** **(%)**
	29.4	0.65–0.87	1.47	0.16–0.24	132.4–237	6–9.8

**Table 2 polymers-14-03650-t002:** Nomenclature adopted for all specimens.

Specimen	Nomenclature
Unaged Epoxy	UE
Water Immersion Epoxy Aged for 240 h	AE/240 h
Water Immersion Epoxy Aged for 720 h	AE/720 h
Water Immersion Epoxy Aged for 1440 h	AE/1440 h
Unaged Composite	UC
Water Immersion Composite Aged for 240 h	AC/240 h
Water Immersion Composite Aged for 720 h	AC/720 h
Water Immersion Composite Aged for 1440 h	AC/1440 h

**Table 3 polymers-14-03650-t003:** DMA parameters for epoxy resin and its composites with 40 vol% fique fabric of both unaged and aged-by-water immersion.

DMA Parameter	UE	AE/240 h	AE/720 h	AE/1440 h	UC	AC/240 h	AC/720 h	AC/1440 h
E${}^{\prime }$ at RT (GPa)	3.21	5.33	5.82	2.26	2.81	1.70	4.14	4.87
End of Glass condition (°C)	58	77	78	74	64	83	84	83
Onset of Rubbery condition (°C)	79	157	158	159	135	158	160	160
E${}^{\prime \prime }$ at RT (GPa)	0.038	0.072	0.082	0.100	0.062	0.119	0.117	0.123
Maximum Internal friction (GPa)	0.434	0.535	0.514	0.500	0.286	0.329	0.283	0.330
Begin glass transition T${}_{g}$ (°C)	68	112	112	110	90	110	113	118
Maximum damping (dimensionless)	0.87	0.64	0.54	0.50	0.40	0.26	0.21	0.21
Dynamic T${}_{g}$	78	129	123	119	87	122	123	119

## Data Availability

The data presented in this study are available on request from the corresponding author.

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
