# Peer review of "Water Immersion Aging of Epoxy Resin and Fique Fabric Composites: Dynamic–Mechanical and Morphological Analysis"

_polymers, 2022, doi:10.3390/polym14173650_

Round 1

Reviewer 1 Report

1) It should be mentioned in the Introduction that Fique fibers are quite a unique natural fiber occuring mainly in Columbia, but the the results of the study and the conclusions are of regional importance and valid for other natural fibers as well.

2) The problem described in the Introduction eventually can be solved by the use of nanosilica in the epoxy resin matrix (subject of a future study?!) - compare rsta.royalsocietypublishing.org Phil. Trans. R. Soc. A 374: 20150275

3) 2.1. Materials: the fibers used to make the fabric need to be described in more detail, i.e. diameter. Is the fabric UD or multiaxial? What is the nature of the stitching fiber (chemistry, diameter, amount used)?

4) 2.2 Drying process and cutting process should be described in much more detail, including the equipment used.

5) 3.2. line 156/157 "...recorded a weight loss in the order of 20 %." This is wrong, as there was no weight loss, but a weight increase of 20 %.

6) 3.2. Figure 4: as you showed water absorption goind into an equilibrium over time (Figure 3), the polynomial fit for the composite is questionable. Please provide more detailed explanation. The linear fit for the pure epoxy is well documented by many studies...

7) It is Tg, not Tg - please correct throughout the paper

Author Response

Manuscript Polymers-1856768

Response to Reviewers

The authors would like to thank the Reviewers for the valuable comments and suggestions that contribute to improve our manuscript. Amendments were provided accordingly and all modifications were marked as Track Changes in the revised version of the manuscript. Responses to each comment, point by point, are given below.

Reviewer #1 comments:

1) It should be mentioned in the Introduction that Fique fibers are quite a unique natural fiber occuring mainly in Columbia, but the the results of the study and the conclusions are of regional importance and valid for other natural fibers as well.

Response:

The authors are grateful for the suggestion to change the Introduction, and are in agreement with the statements made by the Reviewer. The revised version of the article implements the recommendations.

2) The problem described in the Introduction eventually can be solved by the use of nanosilica in the epoxy resin matrix (subject of a future study?!) - compare rsta.royalsocietypublishing.org Phil. Trans. R. Soc. A 374: 20150275

Response:

Indeed, the use of nanosilica in epoxy is the subject of studies parallel to that of degradation, presented in this article. There is the consideration of other nanoparticles as well, however, future results are dependent on the characterization of the behavior presented in this article, once this article presents investigations of the composite considering the epoxy without addition, as well as the natural fiber without surface treatment. Conditions that can be taken as a reference for future research. A note of future research with the addition of nanosilica in the epoxy was added and based on the article "From matrix nano- and micro-phase tougheners to composite macro-properties" (https://doi.org/10.1098/rsta.2015.0275), which is now cited in our revised version.

3) 2.1. Materials: the fibers used to make the fabric need to be described in more detail, i.e. diameter. Is the fabric UD or multiaxial? What is the nature of the stitching fiber (chemistry, diameter, amount used)?

Response:

As recommended by the Reviewer, in the revised version of the article, Table 1 was inserted with the main characteristics of the fiber fiber, as well as the fabric configuration.

4) 2.2 Drying process and cutting process should be described in much more detail, including the equipment used.

Response:

As requested by the Reviewer, the fabric drying processes (surface moisture removal), as well as the composite cutting process and equipment used were described.

5) 3.2. line 156/157 "...recorded a weight loss in the order of 20 %." This is wrong, as there was no weight loss, but a weight increase of 20 %.

Response:

The Reviewer is correct in the observation made. In fact, the correct term to be used was not observed. The error has been corrected and the authors are grateful for the observation.

6) 3.2. Figure 4: as you showed water absorption goind into an equilibrium over time (Figure 3), the polynomial fit for the composite is questionable. Please provide more detailed explanation. The linear fit for the pure epoxy is well documented by many studies...

Response:

As recommended, more detailed explanation is now provided. The Figure 4 shows the linear (top left) and polynomial (bottom left) fit for the epoxy, and the data on the right is for the composite, with the top fit for polynomial and the bottom fit for the linear fit. Based on the R² parameter, the linear fit for the epoxy and the polynomial fit for the composite were plotted, both marked in red boxes. Furthemore, in agreement with the Reviewer's statement, the linear fit is actually chosen to describe water absorption by polymeric materials and polymeric composites, however, due to the polynomial fit being significant for the composite, the authors chose to demonstrate the polynomial fit.

7) It is Tg, not Tg - please correct throughout the paper.

Response:

Complied. The Reviewer is correct and, as recommended, all changes to the text have been made.

Reviewer 2 Report

In this work, the epoxy matrix and the composite, both unaged and aged, were fully characterized by various methods. The main results were that degradation by water present appearance of complex microfibril structures, plasticization of epoxy resin and debonding fique fiber/epoxy matrix. The most intense color change was obtained for the water immersion aged epoxy by 1440h. Cole-Cole diagrams revealed the heterogeneity of the materials studied. The reported results can be helpful both on design and operation stages for epoxy-reinforced fique fabric composites when in contact with water. Accordingly, this work has enough novelty and advance. Thus, I recommend it to be published after minor revisions.

1.     On page 2 Line 67, the epoxy value of the diglycidyl ether of bisphenol A (DGEBA) epoxy resin should been provided.

2.     What do C and D stand for in Figure 3?

3.     The FT-IR of DGEBA/TETA before and after curing should need to be provided to indicate that the epoxy was fully cured.

4.     On page 5 Line 154-157, this part of the description is not clear enough, more discussion should be added in this part. In additional, why there is an increase and then a decrease on weight variation? This result is inconsistent with the water absorption.

5.     Why were the modulus and Tg of unaged resins/composites lower than those of aged resins/composites? This is a strange phenomenon. Could the authors make any comment on this?

Author Response

Manuscript Polymers-1856768

Response to Reviewers

The authors would like to thank the Reviewers for the valuable comments and suggestions that contribute to improve our manuscript. Amendments were provided accordingly and all modifications were marked as Track Changes in the revised version of the manuscript. Responses to each comment, point by point, are given below.

Reviewer #2 comments:

In this work, the epoxy matrix and the composite, both unaged and aged, were fully characterized by various methods. The main results were that degradation by water present appearance of complex microfibril structures, plasticization of epoxy resin and debonding fique fiber/epoxy matrix. The most intense color change was obtained for the water immersion aged epoxy by 1440h. Cole-Cole diagrams revealed the heterogeneity of the materials studied. The reported results can be helpful both on design and operation stages for epoxy-reinforced fique fabric composites when in contact with water. Accordingly, this work has enough novelty and advance. Thus, I recommend it to be published after minor revisions.

Response:

The authors are grateful for the Reviewer's recommendation and acknowledgment of the article, based on the statement that the study is sufficiently novel and advanced.

  1. On page 2 Line 67, the epoxy value of the diglycidyl ether of bisphenol A (DGEBA) epoxy resin should been provided.

Response:

Complied. The Reviewer correctly noted that the description of the resin/hardener ratio was not clear, therefore, in the revised version, the proportion value was clearly provided.

  1. What do C and D stand for in Figure 3?

Response:

The authors imagine that C and “E” are the doubts, so the abbreviations for epoxy (E) and composite (C) were added to the legend, and the use of abbreviations clearer. The authors are grateful for the observation.

  1. The FT-IR of DGEBA/TETA before and after curing should need to be provided to indicate that the epoxy was fully cured.

Response:

Complied. The FTIR data for the epoxy and the composite were inserted, considering the no aging condition and the most severe aging in both materials.

  1. On page 5 Line 154-157, this part of the description is not clear enough, more discussion should be added in this part. In additional, why there is an increase and then a decrease on weight variation? This result is inconsistent with the water absorption.

Response:

As requested by the Reviewer, a greater discussion on composite weight variation was added, aiming to clarify the composite saturation effect. In addition, the data are not at variance with the water absorption, as this considered about 500 hours of immersion, and the weight variation considered up to 1440 hours. When comparing the weight variation values in the 3 different aging times, it is possible to observe that there is no significant variation, both between 20 and 23%, so the observed decrease can be considered an indication of the water saturation of the composite.

  1. Why were the modulus and Tg of unaged resins/composites lower than those of aged resins/composites? This is a strange phenomenon. Could the authors make any comment on this?.

Response:

As presented in the article, the phenomenon is due to the possible presence of bonds between the water molecules and the epoxy molecules (lines 291-294). The following figure is taken from the Panchagnula et al. [46 ] article and details the nature of the water-epoxy interaction has been demonstrated (N: negative end of epoxy chain, P: positive end of epoxy chain, shows hydrogen bonding, O: oxygen and H: hydrogen, blue ring: free water, yellow ring: bound water).

In addition, the essence of the glass transition temperature can be described as the beginning of coordinated molecular movements of polymer chains. At low temperatures, we only have molecular vibration movements and the polymer appears as a rigid glassy material. When the temperature increases, the amplitude of the vibrations increases, progressively reducing the action of the intermolecular forces of interaction. As a consequence the cooperative nature of the vibrations of neighboring atoms is increased. At the glass transition temperature, the chain endings and chain segments acquired enough energy to overcome the intermolecular forces of interaction and presented rotation and translation movements. Tg characterizes the beginning of these movements and especially of chain segments with 20 to 50 consecutive carbon atoms. The rotational and translational motion modes work as an important energy absorption mechanism. The glass transition is not a 1st order transition from a thermodynamic point of view, once there is no discontinuity in entropy or in the free volume of the polymer as a function of temperature. When the vibration energy (thermal) exceeds the binding energy of the atoms, the polymer decomposes (breaking of covalent bonds). Indeed, the displacement of the curves for higher temperatures and lower tan delta values was quite significant, however this fact indicates the clear interference of water in the epoxy matrix and in its curing process.
